# Establishment and characterization of VOA1066 cells: An undifferentiated endometrial carcinoma cell line

Yemin Wang[1,2]*, Valerie Lan Tao[1], Chae Young Shin[2], Clara Salamanca[2], Shary Yuting Chen[2], Christine Chow[3], Martin Köbel[4], Susana Ben-Neriah[5], David Farnell[2], Christian Steidl[2,5], Jessica N. Mcalpine[6], C. Blake Gilks[2], David G. Huntsman[1,2,6]*

**1** Department of Molecular Oncology, British Columbia Cancer Research Institute, Vancouver, BC, Canada, **2** Department of Pathology and Laboratory Medicine, University of British Columbia, Vancouver, BC, Canada, **3** Genetic Pathology Evaluation Center, Vancouver General Hospital and University of British Columbia, Vancouver, BC, Canada, **4** Department of Pathology and Laboratory Medicine, University of Calgary, Calgary, AB, Canada, **5** Department of Lymphoid Cancer, British Columbia Cancer Research Institute, Vancouver, BC, Canada, **6** Department of Obstetrics and Gynaecology, University of British Columbia, Vancouver, BC, Canada

* yewang@bccrc.ca (YW); dhuntsma@bccancer.bc.ca (DGH)

**Data Availability Statement:** The whole exome sequencing data are available at the National Centre for Biotechnology Information (NCBI) database (SRA accession: PRJNA645500).

## Abstract

Dedifferentiated endometrial carcinoma (DDEC) is a rare but highly aggressive type of endometrial cancer, in which an undifferentiated carcinoma arises from a low-grade endometrioid endometrial carcinoma. The low-grade component is often eclipsed, likely due to an outgrowth of the undifferentiated component, and the tumor may appear as a pure undifferentiated endometrial carcinoma (UEC). We and others have recently identified inactivating mutations of SMARCA4, SMARCB1 or ARID1B, subunits of the SWI/SNF chromatin-remodeling complex, that are unique to the undifferentiated component and are present in a large portion of DDEC and UEC. However, the understanding of whether and how these mutations drive cancer progression and histologic dedifferentiation is hindered by lack of cell line models of DDEC or UEC. Here, we established the first UEC cell line, VOA1066, which is highly tumorigenic *in vivo*. This cell line has a stable genome with very few somatic mutations, which do include inactivating mutations of *ARID1A* and *ARID1B* (2 mutations each), and a heterozygous hotspot *DICER1* mutation in its RNase IIIb domain. Immunohistochemistry staining confirmed the loss of ARID1B, but ARID1A staining was retained due to the presence of a truncating non-functional ARID1A protein. The heterozygous *DICER1* hotspot mutation has little effect on microRNA biogenesis. No additional *DICER1* hotspot mutations have been identified in a cohort of 33 primary tumors. Therefore, we have established the first UEC cell line with dual inactivation of both ARID1A and ARID1B as the main genomic feature. This cell line will be useful for studying the roles of ARID1A and ARID1B mutations in the development of UEC.

**Funding:** This work was supported by research funds from the National Cancer Institute of NIH (United States) (1R01CA195670-01, https://www.cancer.gov, to DGH), the Terry Fox Research Institute Initiative New Frontiers Program in Cancer (TFF1021, https://www.tfri.ca, to DGH), and the Carraresi Foundation and the Sumiko Kobayashi Marks Memorial OVCARE Research Grants supported by the VGH & UBC Hospital Foundation (2019 competition, http://www.ovcare.ca, to YW). The funders had no role in study design, data collection and analysis, decision to publish, or preparation of the manuscript.

**Competing interests:** The authors have declared that no competing interests exist.

## Introduction

Endometrial cancer (EC) is the most common form of gynecological cancers found worldwide and can be categorized into 2 types: an estrogen-dependent type I that pertains to the majority of EC and an estrogen-independent type II, which presents as a more aggressive form of EC [1]. The majority of type I ECs are classified as endometrial endometrioid carcinomas, which are associated with a lower FIGO stage and have better prognosis. In contrast, type II ECs have non-endometrioid histology, as cells morph into quick-dividing, high grade cancers that are associated with poor prognosis. One subtype of type II EC is dedifferentiated endometrial carcinoma (DDEC), in which an undifferentiated carcinoma arises from a low-grade endometrial endometrioid carcinoma that is often a mismatch repair deficient (MMRd) tumor [2–4]. The low-grade component is eclipsed in about 40% cases, likely due to an outgrowth of the undifferentiated component, and the tumor overall appears as a pure undifferentiated endometrial carcinoma (UEC) [2, 5]. The undifferentiated component shows sheet-like proliferation of monomorphic round to polygonal-shaped cells lacking both cellular cohesion and evidence of epithelial architecture (absence of glands, nests or trabeculae) [6]. The endometrioid components usually express estrogen receptor (ER) and PAX8, whereas the undifferentiated components are negative for both markers.

DDEC occurs in women with a peak of age of diagnosis at 55 years and is clinically aggressive. Most patients quickly succumb to their diseases even when the undifferentiated component appears as a minor fraction of the tumor [2, 4]. Clinically, patients are treated with the same regimen that is used for other high-risk EC, which depending on the institution includes external beam pelvic radiotherapy +/- chemotherapy with carboplatin/paclitaxel. DDEC has been reported to have a worse prognosis compared to grade 3 endometrioid EC with overall survival rates ranging from 41–75% and median survival time of just 6 months [7]. There is an urgent need to understand the biology of this disease and to improve clinical management and outcomes. Such efforts have been hindered by the lack of cell line models for DDEC or UEC.

In this study, we have generated the first UEC cell line from a patient's primary tumor specimen. We have characterized the molecular feature of this cell line, which will be an important preclinical tool to better understand the pathogenesis of the disease, and to investigate the effectiveness and efficacy of various preclinical drugs. Ultimately, this cell line has the potential to contribute to the progression of better treatment plans and to clinical outcomes for UEC patients.

## Materials and methods

### Patient cohort

This study included 34 endometrial cancer cases that were obtained through the OVCARE tumor bank at the Vancouver General Hospital, British Columbia, Canada, with the ethics being approved by the University of British Columbia Institutional Review Board. All patients have provided written consents. The VOA1066 cell line was derived from the endometrial tumor tissue of a 74-year old woman (BMI 24.7) who presented with pain and vaginal bleeding. The patient underwent an endometrial biopsy that revealed an undifferentiated carcinoma with differential diagnosis including endometrial stromal sarcoma and carcinosarcoma. An ultrasound scan revealed a thickened endometrial echo of 3.1 cm. During surgery, her uterus was markedly enlarged, had gross disease in her omentum, and had suspicious retroperitoneal nodes. Given the nature of the disease, she underwent a hysterectomy, bilateral salpingoophorectomy, omentectomy and lymph node debulking with residual disease at the completion of surgery and on subsequent imaging. Final pathology review revealed extensive myometrial

invasion by an undifferentiated carcinoma, extending almost to the uterine serosa, with extensive lymph vascular space invasion and involvement of the right fallopian tube, right ovary, omentum, and right pelvic lymph node. Through her recovery, she was unwell and was unable to receive chemotherapy prior to further progression of the disease. Ultimately, she succumbed to her disease 6 weeks postoperatively.

## Cell line establishment and culture

To establish the VOA1066 cell line, the primary tumor tissue was collected at the time of surgery after obtaining an informed consent from the patient. For cell line derivation, the tumor tissue was minced and digested using 0.5% Trypsin/EDTA. After 75 minutes, single cells from the solution were placed in two 100 mm dishes in 199:105 media fortified with 10% defined FBS (Hyclone) and 50 ug/mL of Gentamicin (Thermofisher). Cells were left to adhere and grow at 37 degrees in a 5% $CO_2$ incubator for 7 days before being passaged. Culture medium was replaced every 3–4 days and the cancer cells were enriched by differential trypsinization and subsequent removal of fibroblasts. Cells at 80% confluency were split in the ratio of 1:4 twice a week. The cells were negative for mycoplasma (IDEXX BioResearch Case # 12483–2017). The short tandem repeat (STR) analysis was carried out in DNA from cell culture and fresh frozen tissue from the patient by Genetica. The high-grade endometrial carcinoma cell line HEC50 was obtained from the JCRB cell bank, STR verified and maintained in RPMI plus 10% characterized FBS (Hyclone) at 37 degrees in a 5% $CO_2$ incubator.

## Karyotyping

VOA1066 cell metaphases were prepared according to standard cytogenetic procedures. Metaphase capturing was performed by an automated hardware and software system (Metafer MSearch Metaphase Finder) by Metasystems, Altlussheim, Germany. Captured metaphases (n = 20) were selected randomly for complete G-banding analysis (band to band).

## Xenograft tumor growth

This study was approved by the animal care committee of the University of British Columbia (A17-0146). VOA1066 cells were inoculated subcutaneously (n = 4) into the back or intraperitoneally (n = 6) into female NRG mice, obtained from British Columbia Cancer Research Institute Animal Research Centre, with a total of 5 million cells per mice. Mice, in groups of 3 or 4 per cage, were housed in ventilated cages with free access to water and low-fat food, nesting materials and plastic hiding structures "huts" for additional environmental enrichment. Mice were anaesthetized prior to tumor implantation to reduce distress and suffering. Hydrogel and dough diets were provided after tumor inoculation. For subcutaneous tumors, tumour growth was monitored twice a week using callipers to determine the length (mm) and width (mm) of the tumour, the volume of the tumour (L x $W^2$ x 0.52) was calculated and the mouse was weighed until the experimental endpoint of 1500 $mm^3$ or humane endpoints was reached. For the intraperitoneal model, mice were closely monitored for illness as a sign of tumor development until the humane endpoints. Evaluation for clincal signs includes the following catergories: dehydration, abnormal gait or reduced movement/activity, abdominal distension, grooming/posture, respiratory changes and body weight loss. No mortality was observed in this study before humane endpoints. At the experimental endpoints, all mice were euthanized by inhalant anesthetic (isofluroane) followed by carbon dioxide. Tumour masses were collected and weighed. Tumour tissues were fixed in 10% neutral buffered formalin, embedded or sectioned for histology analyses.

## Immunohistochemistry (IHC)

IHC was performed on whole slides of VOA1066 primary tumor, adjacent normal endocervix tissue and cell line-derived xenografted tumors using anti-ARID1A (clone EPR13501, ab182560, Abcam), anti-ARID1B (clone 2D2, H00057492-M01, Abnova), anti-ER (clone EP1, GA084, Dako), anti-cytokeratin 7 (OV-TL12/30, GA619, Dako), anti-EMA (E29, IR629, Dako) and anti-PAX8 (MRQ-50, 363M-16, Cell Marque) as previously described [8, 9]. Staining was evaluated by gynecologic pathologists (D.F., M.K.).

## Cell morphology and growth curve

VOA1066 cells were seeded in DMEM, RPMI, and 199:105 supplemented with 10% FBS in 96-well plates (Sarstedt) in pentaplicates. Phase contrast images were taken every 6 hours for 6 consecutive days using an Incucyte Zoom live-cell imaging and analysis system. The confluency of VOA1066 cells culturing in individual media were obtained and plotted to generate the growth curve.

## DNA isolation

VOA1066 cells and the fresh frozen tumour from the original patient were processed for DNA extraction following the DNeasy Blood & Tissue Protocol from Qiagen. The FFPE block of tumor adjacent benign tissues from the same patient was sectioned and extracted using the QIAamp DNA FFPE Tissue Protocol.

## Whole exome sequencing and Sanger sequencing analyses

A total of 1.0μg genomic DNA per sample was used as input material for the DNA library preparation and whole exome sequencing at Novogene (Sacramento, CA). Briefly, sequencing libraries were generated using Agilent SureSelect Human All Exon kit (Agilent Technologies, CA, USA) following manufacturer's recommendations, and index codes were added to each sample. Products were purified using AMPure XP system (Beckman Coulter, Beverly, USA), quantified using the Agilent high sensitivity DNA assay on the Agilent Bioanalyzer 2100 system and then sequenced on an Illumina HiSeq2500 platform. Burrows-Wheeler Aligner (BWA) was utilized to map the paired-end clean reads to the human reference genome (b37+-decoy, ftp://gsapubftp-anonymous@ftp.broadinstitute.org/bundle/b37/human_g1k_v37_decoy.fasta.gz). SAMtools was used for sorting the BAM file, and Picard was utilized to mark duplicate reads. The sequencing depth was 58.48, 103.81 and 86.99 for adjacent benign tissue (FFPE), primary tumor and cells of VOA1066, respectively. The tumor mutation burden, mutation per Mb, was calculated by dividing the number of non-synonomous SNP and indel to the total sequencing coverage. The whole exome sequencing data are available at the National Centre for Biotechnology Information (NCBI) database (SRA accession: PRJNA645500). The *DICER1* RNase IIIb domain mutation was further validated using Sanger sequencing as previously described [10].

## Western blotting

A pellet of VOA1066 cells was lysed in urea buffer that contains 90% 8.8 M urea, 2% 5 M $NaH_2PO_4$, 8% 1M Tris pH 8.0 with freshly added 10x Protease and 100x phosphatase inhibitor. 30 ug of cell lysate was resolved on a 6% SDS-PAGE gel for protein detection as previously described [11]. Primary antibodies include ARID1A (Sigma, HPA005456, 1:1000), ARID1B (Abgent, AT1190a, 1:2000), SMARCA2 (Cell Signaling, #6889), SMARCA4 (Abcam, ab110641, 1:2000), SMARCB1 (#612110, BD), p-AKT (S473) (Cell Signaling, #9271), total

AKT (Cell Signaling, #9272) and PTEN (Cell Signaling, #9552S) and Vinculin (Sigma, v9131, 1:50000).

## Co-immunoprecipitation

Cells were lysed in Buffer A (20 mM HEPES pH 7.9, 10 mM KCl, 0.2 mM EDTA pH 8, 1X Halt™ protease inhibitor cocktail) on ice for 10 minutes. After addition of 10% NP-40, extracts were vortexed, and nuclei were pelleted by centrifugation at 21,000 RCF at 4 degree for 30 seconds. Nuclei were resuspended in Buffer C (20 mM HEPES pH 7.9, 0.4 M NaCl, 10 mM EDTA pH 8, 1 mM EGTA pH 8, 1X Halt™ protease inhibitor cocktail) and incubated at 4 degree for 15 minutes, followed by centrifugation at 21,000 RCF at 4 degree for 5 minutes. Soluble nuclear protein extracts were collected and protein concentration was measured using a BCA microplate assay. An input of 500 μg of nuclear protein was used for immunoprecipitation using normal IgG or anti-SMARCC1 (Cell Signaling, #11956, 4ug per sample) antibody in a total volume of 500 μL. 30 μL of a 50% slurry of Protein A conjugated sepharose beads were added to each sample followed by incubation at 4 degree overnight using end-over-end rotation. Sepharose beads were pelleted by centrifugation at 6,000 RCF at 4 degree for 1 minute and IP samples were washed 5 times with 500 μL Buffer C before being processed for western blotting analysis.

## RNA extraction and microRNA expression analysis

RNA was extracted using the miRNeasy kit (Qiagen) according to the manufacturer's protocols. Individual miRNAs were quantified using Exiqon's PCR system following the manufacturer's instructions as previously described [12]. The levels of miRNAs were assayed in triplicate and the Ct values were normalized to the endogenous small nuclear RNA RNU1A1 to obtain dCt values.

## Statistical analysis

The data are summarized as means ± standard deviation (SD). Student's *t*-test was used to evaluate the significant difference between two groups in all experiments. Calculations and analyses were performed using GraphPad Prism 6 software (GraphPad Software, La Jolla, CA). A *p*-value < 0.05 was considered significant.

# Results

## IHC features of the VOA1066 primary tumor

To validate the diagnosis of VOA1066 primary tumor, we performed IHC stainings on both the primary tumor speciemen of VOA1066 case and its adjacent normal endocervix tissue to determine the expression status of multiple Mullerian epithelial markers. It revealed that the VOA1066 primary tumor cells lost the expression of four epithelial markers including ER, cytokeratin-7 (CK7), EMA and PAX8, which were retained in the epithelial cells of the adjacent normal endocervix tissue (**Fig 1**).

## Cell morphology and proliferation

The VOA1066 cell line has stably self-immortalized and currently surpassed 50 passages. The STR profile of the VOA1066 cell line was a 100% match to the original VOA1066 tumor (**Table 1**), which did not match to any other cell lines in the publicaly available STR databases for cell lines.

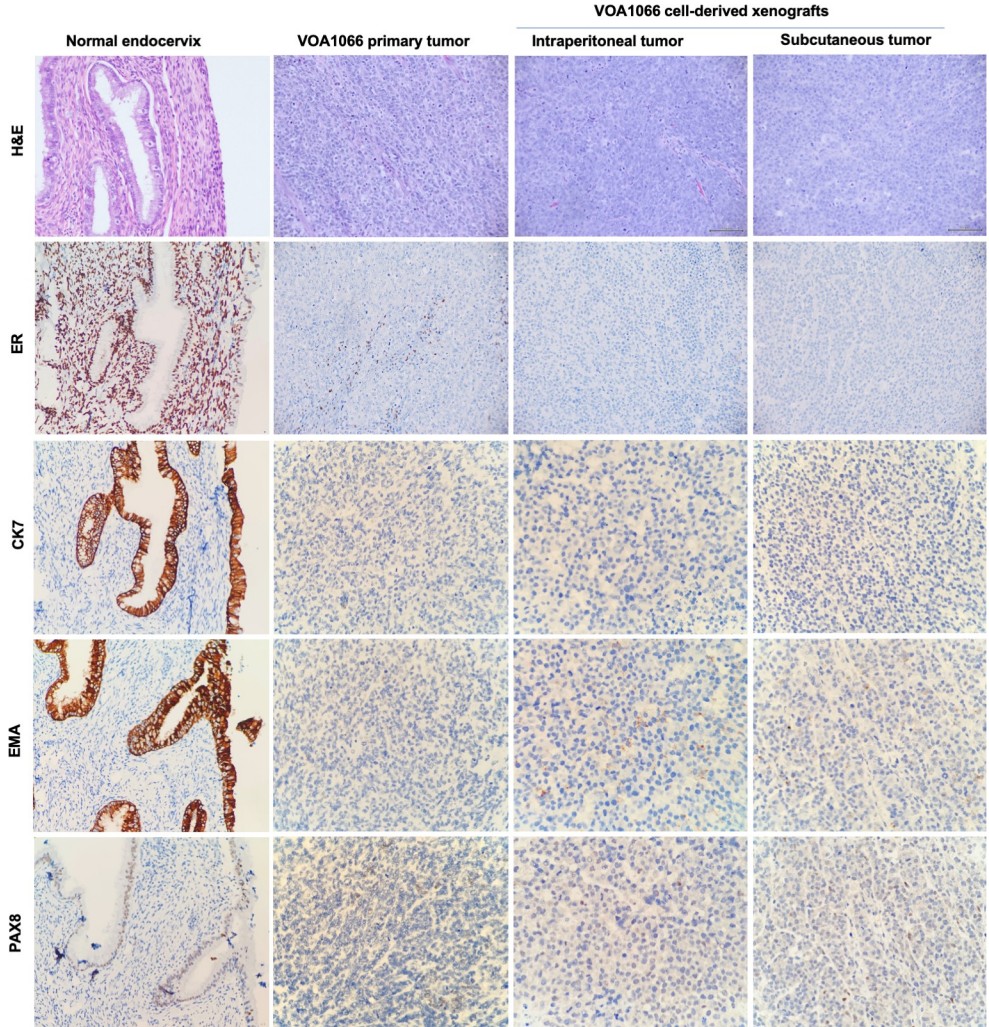

**Fig 1. H&E and ER IHC staining of VOA1066 samples.** Both the primary tumor and xenografted subcutaneous and intraperitoneal tumors of VOA1066 showed minimal focal or no staining of Mullerian epithelial markers ER, CK7, EMA and PAX8.

A 6-day growth curve was established using the IncuCyte Zoom, a live-cell imaging and analysis system, to compare the morphology and proliferation rate of VOA1066 cells in various media types such as 199:105, DMEM and RPMI. The cell morphology of VOA1066 cells was consistent across all media types, displaying cobblestone-like patterns with a fibroblast-like cell shape (Fig 2A). The cells remained morphologically identical in late passages. VOA1066 cells proliferate best in DMEM media fortified with 10% FBS with a doubling time of ~35 hours (Fig 2B). Thus, all following experiments of VOA1066 cell line were cultured in DMEM media. Noteworthily, this cell line tends to aggregate together and form spheroids spontaneously if cells are not well-dissociated during passaging.

### *In vivo* tumor growth in mice

To evaluate the growth potential of VOA1066 cells in vivo, we inoculated the cells into immunodeficient mice either subcutaneously or intraperitoneally. The subcutaneous tumors developed in all four mice with measurable tumors 12 days post inoculation, which grew rapidly

**Table 1. STR analysis results of the parental tumor and VOA1066 cell line.**

| STR loci | Primary tumor | Cell line |
| --- | --- | --- |
| STR Loci | Alleles | Alleles |
| D5S818 | 11, 15 | 11, 15 |
| D13S317 | 11, 12 | 11, 12 |
| D7S820 | 8, 11 | 8, 11 |
| D16S539 | 11, 12 | 11, 12 |
| vWA | 16 | 16 |
| TH01 | 7, 9.3 | 7, 9.3 |
| AMEL | X | X |
| TPOX | 9, 11 | 9, 11 |
| CSF1PO | 10, 11 | 10, 11 |
| D3S1358 | 14, 15 | 14, 15 |
| D21S11 | 29, 31.2 | 29, 31.2 |
| D18S51 | 12, 17 | 12, 17 |
| Penta E | 7, 10 | 7, 10 |
| Penta D | 12 | 12 |
| D8S1179 | 11, 13 | 11, 13 |
| FGA | 23, 24 | 23, 24 |

and approached 1500 mm$^3$ in size within 2 weeks after being detectable (**Fig 2C**). The intraperitoneal tumors also developed quickly, as indicated by rapid gain of body weight and mild signs of illness a week post inoculation, which progressed to humane endpoints in four more weeks. Upon termination, gross examination indicated that all 6 mice developed intraperitoneal tumors, among which four invaded into the peritoneal wall, and all of them developed distant metastasis into liver (n = 4), stomach/intestine (n = 4), ovary (n = 4), kidney (n = 2) and/or pancreas (n = 2) (**Fig 2D**). Histologic examination confirmed that both subcutaneous and intraperitoneal tumors were morphologically similar to that of the parental tumor and did not express estrogen receptor alpha (ER) by IHC, a marker for differentiated endometrial endometrioid carcinoma that was negative in the primary tumor (**Fig 1**).

## Chromosome analysis

In order to determine whether the VOA1066 cells display chromosomal instability, we analyzed 20 metaphases. The number of chromosomes was 46, XX with a single translocation [t (3;14)(p25;q21)] identified (**Fig 3**), suggesting that the VOA1066 cell line is a chromosomal stable cell line.

## Genomic characterization

To further characterize the genomic alterations in the VOA1066 cell line, we performed whole exome sequencing for VOA1066 cell line, the primary tumor and its adjacent benign endometrial tissue. The tumor mutation burdens of the primary tumor and VOA1066 cell line were 0.011 and 0.013 mutations per Mb, suggesting that they are genome stable. A total of 68 and 54 somatic exonic SNVs were identified in cell line and the primary tumor, respectively. This included 16 and 15 synonymous variants as well as 52 and 39 protein-altering mutations for cell line and primary tumor, respectively. Among the non-synonymous mutations, 33 were present in both cell line and primary tumor, including truncating mutations of *ARID1A* and *ARID1B* genes and a cancer hotspot missense mutation of *DICER1* gene (p.D1810Y).

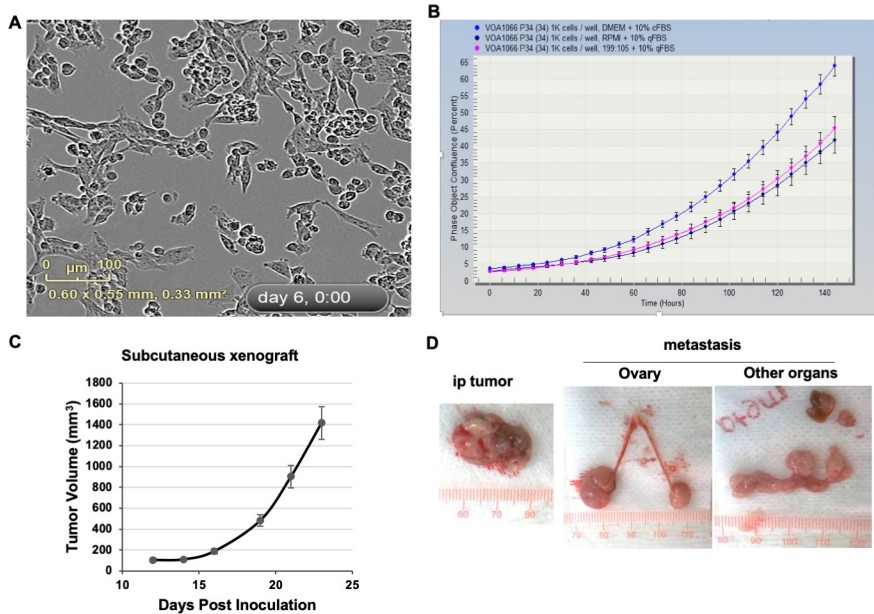

**Fig 2. The proliferation and morphology of VOA1066 cells *in vitro* and *in vivo*.** (**A**) A representative image of VOA1066 cells on petri dish. (**B**) Growth curves of VOA1066 in different culture media. (**C**) Subcutaneous tumor growth curve of VOA1066 cells. (**D**) Representative images of intraperitoneal xenografted tumors and their metastasis of VOA1066 cells.

Furthermore, there are only 3 and 2 exonic small deletions or insertions in cell line and primary tumor, respectively, including small insertions in both *ARID1A* and *ARID1B* genes (**Fig 4A**). The average allelic depth for common non-synonymous mutations was 22, while that for non-synomymous mutations unique to either VOA1066 cell line or primary tumor was 6 (**S1 Table**), suggesting that the exome sequencing depth may not be sufficient to detect these rare events in one of the samples.

Due to the fact that we have previously discovered co-current mutations and protein loss of ARID1A and ARID1B, two mutually exclusive accessory components of the SWI/SNF BAF chromatin-remodeling complex [6], in DDEC and UEC [9, 13], we determined the protein

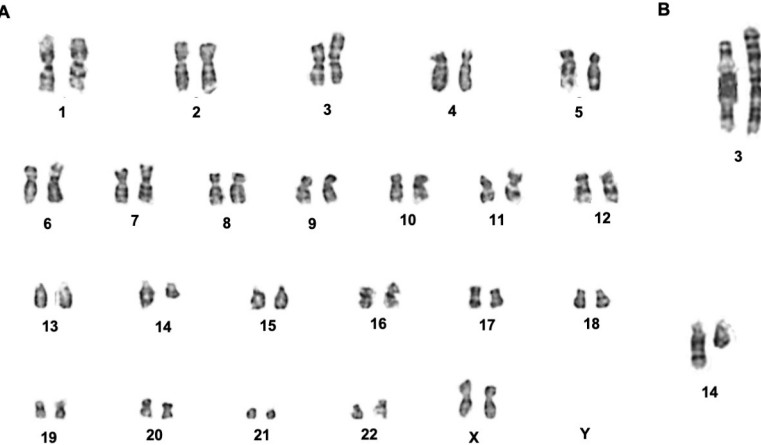

**Fig 3. Karyotyping of VOA1066 cells.** Analysis of 20 metaphases of VOA1066 cells revealed that VOA1066 cells have a stable diploid genome (**A**) with a translocation between chromosomes 3 and 14: 46,XX, t(3;14)(p25;q21) (**B**).

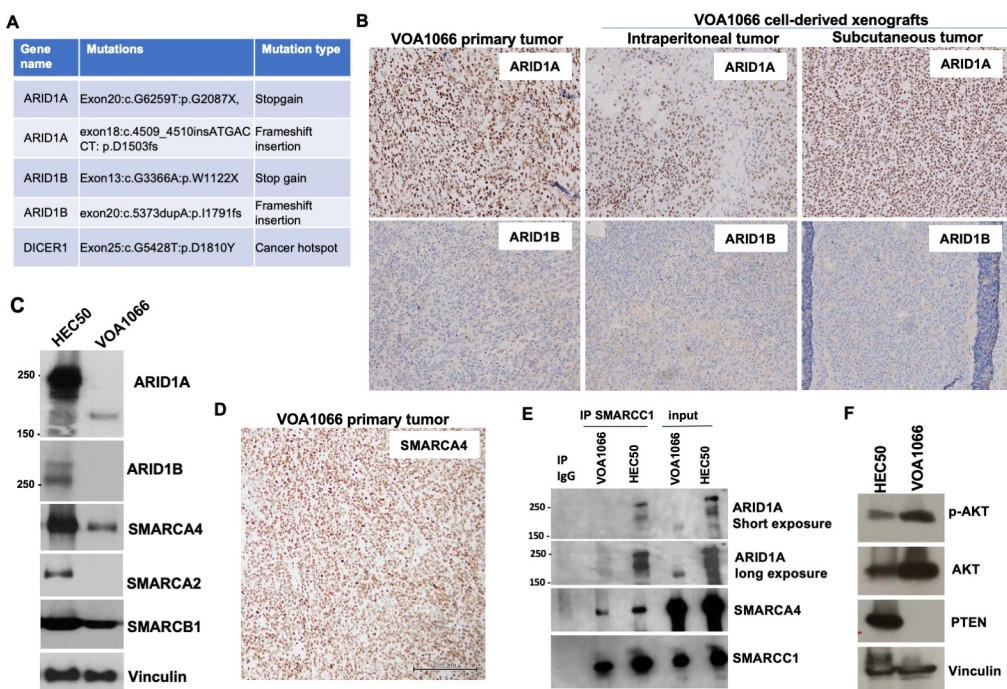

**Fig 4. Mutations and protein expression of *ARID1A* and *ARID1B* genes in VOA1066.** (**A**) Whole exome sequencing revealed shared somatic mutations between primary tumor and cell line, including two inactivating mutations of ARID1A, two inactivating mutations of ARID1B and a RNase IIIb domain hotspot mutation of DICER1. (**B**) IHC staining of ARID1A and ARID1B VOA1066 samples. (**C**) Western blotting analysis of SWI/SNF proteins in VOA1066 and HEC50 cells. (**D**) Representative image of SMARCA4 staining in VOA1066 primary tumor. (**E**) Co-immunoprecipitation analysis of SWI/SNF protein interactions in VOA1066 and HEC50 cells. (**F**) Western blotting analysis of PTEN and pAKT-473 levels in VOA1066 and HEC50 cells.

expression of ARID1A and ARID1B in both primary tumor and xenograft tumors by IHC. All tumor samples had complete loss of ARID1B protein expression, but retained ARID1A expression (**Fig 4B**). Western blotting analysis revealed that the VOA1066 cells expressed neither ARID1B nor the full-length ARID1A protein (>250 kD), but expressed a truncated protein at the size of about 200 kD (**Fig 4C**). In addition, this cell line had intact expression of both SMARCA4 and SMARCB1, two SWI/SNF proteins whose loss is mutually exclusive to ARID1A/ARID1B dual loss in DDEC and UEC [13], but did not express SMARCA2, the ATPase of the SWI/SNF complex that is an alternate to SMARCA4, and known to be frequently lost in DDEC along with SMARCA4 [14]. IHC confirmed that the expression of SMARCA4 was intact in the primary tumor of VOA1066 (**Fig 4E**). To address whether the truncated ARID1A protein is incorporated into SWI/SNF complexes in VOA1066 cells, we performed co-immunoprecipitation using an antibody against SMARCC1, the core subunit of the SWI/SNF complex. Our data revealed that while SMARCA4 is co-immunoprecipitated with SMARCC1 in both cell lines, only the full length ARID1A protein in HEC50 cells, but not the truncated ARID1A protein in VOA1066 cells, is co-immunoprecipitated with SMARCC1 (**Fig 4E**). This finding strongly supports that VOA1066 cell line is defective for both ARID1A and ARID1B. Furthermore, the western blotting analysis demonstrated that VOA1066 cells also lost the expression of PTEN and had a high level of pAKT, indicating the activation of the PI3K/AKT pathway (**Fig 4E**), which occurs frequently in EC, including DDEC [3, 15, 16].

### *DICER1* RNase IIIb domain hotspot mutations in DDEC

To validate that p.D1810Y mutation in VOA1066 samples, we performed Sanger sequencing using DNA samples of VOA1066 cell line, primary tumor and the matched benign tissue adjacent to the primary tumor. Sanger sequencing confirmed that the p.D1810Y mutation occurred as a heterozygous somatic event in both primary tumor and the derived cell line (**Fig 5A**). In addition, through analyzing the RNA of VOA1066 cells by Sanger sequencing, we further confirmed that the heterozygous p.D1810Y mutation is present in the *DICER1* transcript of VOA1066 cells (**Fig 5A**). Quantitation of the levels of several microRNAs showed that 5p strand-derived microRNAs, whose production is severely impaired by *DICER1* RNase IIIb mutations [12, 17], expressed at comparable levels as a *DICER1*-wildtype cell line (**Fig 5B**), suggesting that VOA1066 cells maintain the function of wildtype *DICER1*. Furthermore, when screening a cohort of 33 additional DDEC cases, no *DICER1* RNase IIIb domain hotspot mutation was identified (**Fig 5C**).

We have previously reported *DICER1* RNase IIIb domain hotspot mutations in a minor portion of EC [10]. IHC staining of 6 EC cases (5 endometrioid carcinoma and 1 carcinosarcoma) with *DICER1* hotspot mutations [10] only identified ARID1A loss in one endometrioid carcinoma case, but neither SMARCA4 nor ARID1B was lost in any of the 6 cases. In The Cancer Genome Atlas (TCGA) EC cohort, 17 out of 718 patients have *DICER1* hotspot mutations in their tumors (www.cbioportal.org). Eight of these 17 tumors has inactivating mutations in

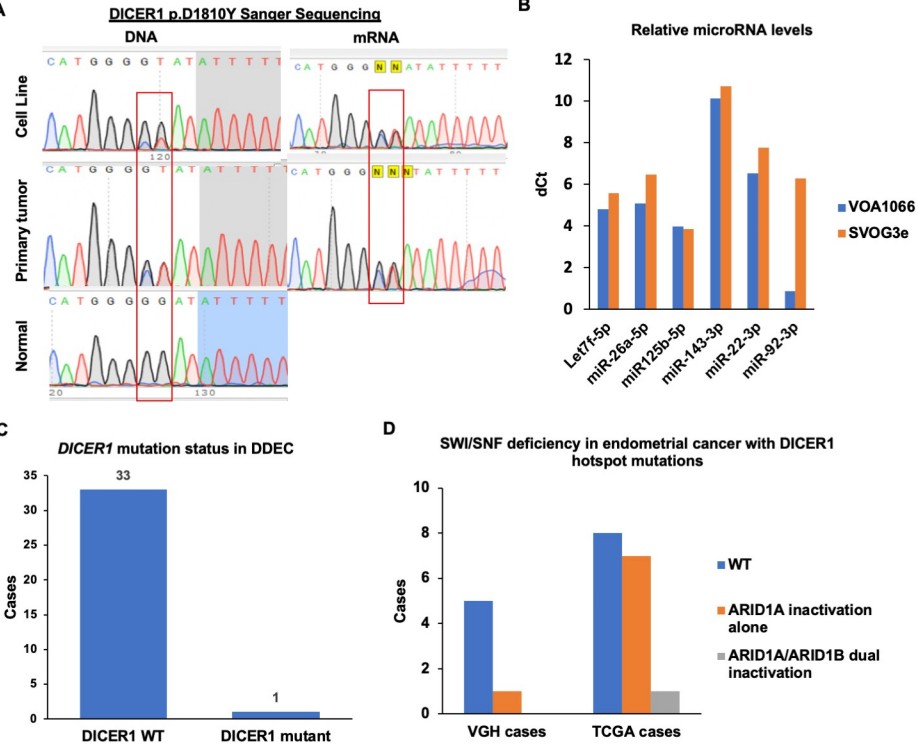

**Fig 5. Mutations of *DICER1* genes in DDEC.** (**A**) Sanger sequencing validated the *DICER1* p.D1810Y mutation in both DNA and RNA samples of VOA1066 cells. (**B**) MicroRNA analyses in VOA1066 cells and SVOG3e cells, a DICER1-wildtype granulosa cell line. (**C**) Mutational analysis of *DICER1* RNase IIIb domain in additional 33 DDEC cases. (**D**) IHC analysis of SWI/SNF proteins (SMARCA4, SMARCB1, ARID1A, ARID1B) in 6 cases of endometrial cancer with *DICER1* RNase IIIb domain hotspot mutations at Vancouver General Hospital (VGH) and mutational analyses of 17 cases of TCGA endometrial cancers with *DICER1* RNase IIIb domain hotspot mutations.

*ARID1A*, but only one case has inactivating mutation in *ARID1A*, which is concordant to *ARID1A* inactivating mutation (**Fig 5D**).

## Discussion

Faithful cell lines representative of tumors are essential for understanding cancer biology and developing putative therapeutic strategies for cancer management. Although many EC cell lines have been established and are available in major cell depositories, no validated cell lines of DDEC or UEC has ever been reported. This is particularly challenging in tumors such as DDEC with diverse components, where only one component may be sampled for pathology analyses or cell line derivation. In this study, we established a cell line, named VOA1066, from a histologically well characterized UEC specimen that was MMR intact and negative for ER. This cell line is chromosomal and genome stable, with a single translocation identified by karyotype, defects in both ARID1A and ARID1B proteins, and a hotspot mutation in RNase IIIb domain of DICER1 identified by whole exome sequencing. It is highly tumorigenic with great metastatic ability in immunodeficient mice. Xenografted tumors morphologically phenocopied the undifferentiated nature of the primary tumor.

Although the genomic landscapes of DDEC are not fully characterized, we and others have recently revealed three types of inactivating mutations along with protein loss occurring in subunits of the SWI/SNF chromatin remodeling complex [6, 9, 13, 18, 19]: SMARCA4, SMARCB1 or ARID1B. Loss of SMARCA4 or SMARCB1 is restricted to the undifferentiated component of DDECs. While loss of ARID1B is also restricted to the undifferentiated component of DDECs, it occurs concordantly with ARID1A loss that usually occurs in both low-grade and undifferentiated components [14, 20]. Furthermore, mutational loss of *SMARCA4*, *SMARCB1* or *ARID1A/ARID1B* is often associated with loss of SMARCA2 protein, the alternative ATPase of the SWI/SNF complex, through non-genetic mechanisms [14, 20, 21]. Although inactivating mutations were only seen in *ARID1A* and *ARID1B* genes, IHC only confirmed loss of ARID1B in VOA1066 primary tumor and cell line. The retained staining of ARID1A is likely due to a truncated form of ARID1A as a result of the p.D1503fs mutation and/or the p.G2087X mutation in the last exon, which can be still recognized by both ARID1A antibodies (immunogens: 1200–1350 aa (Abcam, IHC) and 1266–1370 aa (Sigma, western blotting)). The C-terminus of ARID1A and ARID1B proteins have been shown to be essential for their interaction with SMARCA4 or SMARCA2 and for interaction with nuclear receptors [22]. Accordingly, our co-immunoprecipitation confirmed that this C-terminus truncated ARID1A was not be incorporated into SWI/SNF complexes and thus lose the SWI/SNF-dependent activity. Since the diagnostic value of ARID1A IHC has been proposed [23], this study raised the concern that some frameshift or nonsense mutations may generate a truncated non-functional form of ARID1A that can be recognized by ARID1A antibodies not recognizing the C-terminus, leading to false positive detection of functional ARID1A proteins.

Another surprising finding is that this cell line, as well as the primary tumor, carries a missense hotspot mutation of *DICER1* in its RNase IIIb domain. We have previously reported *DICER1* RNase IIIb mutations in sex-cord stromal tumors of the ovary [12, 24] and in about 0.6% of EC [10]. The RNase IIIb mutation of *DICER1* usually occurs along with inactivation of the other allele, resulting complete loss of the RNase IIIb activity of DICER1 and subsequent biased maturation of microRNAs from their precursors. However, the impact of the *DICER1* mutation in VOA1066 cells appears to be low, which is supportd by three facts: 1) we did not identify the second hit *DICER1* mutation; 2) both wildtype and mutant alleles were present in *DICER1* transcripts, and 3) the levels of 5p-strand-derived microRNAs did not differ between VOA1066 cells and a *DICER1*-wildtype cell line. Furthermore, we have also shown that

*DICER1* RNase IIIb domain mutations were not present in additional 33 DDEC cases, and a survey of 23 EC cases with *DICER1* hotspot mutations (6 local cases and 17 cases from TCGA) identified inactivating mutations in both *ARID1A* and *ARID1B* in only one case that is not diagnosed as DDEC in the TCGA database. Therefore, the *DICER1* mutation is likely an independent event from the SWI/SNF mutations that is very rare in DDEC, possibly a passenger event.

Clinical behaviour of DDEC is usually very aggressive and response rates to conventional chemotherapy typically poor. DDEC can be any of the four molecular subtypes of EC [25–27] i.e. POLEmut (with a mutation in the exonuclease domain of polymeras epsilon (*POLE*) leading to an ultramutated tumor), MMRd (mismatch repair insufficiency, associated with hypermutated EC), p53abn (serous-like and corresponding to tumors with high numbers of somatic copy number alterations in TCGA), and NSMP (no specific molecular phenotype, lacking *POLE* mutation, MMR deficiency or *TP53* mutations). These account for 12%, 44%, 19% and 25% of DDEC/UEC, respectively [25, 26]. Furthermore, DDEC/UEC with a mutation in POLE have a more favorable prognosis, suggesting that the prognostic significance of the molecular subtypes may also be applicable within this subset of EC, and have to be taken into account when planning treatment [26]. The VOA1066 cell line is an example of NSMP EC, lacking *POLE* mutation, MMR deficiency or *TP53* mutation. Concordant with the worse outcome associated with the SWI/SNF deficiency in DDEC, the patient of which VOA1066 was derived from succumbed to the disease due to rapid progression of the disease before any conventional therapy. However, since the SWI/SNF deficiency occurs in both MMRd and MMR-intact DDEC, it is unclear whether the TCGA classification or the integrity of the SWI/SNF complex, or their combination, holds the most robust prognostic value in DDEC. Nevertheless, the VOA1066 cell line represents a genomic stable and highly aggressive group of DDEC.

Taken all together, our present study provides the first histologically validated and molecularly characterized UEC cell line that has a very stable genome with dual deficiency of ARID1A and ARID1B as the key genetic feature. This opens the opportunity to evaluate the biological function of SWI/SNF mutations, particularly ARID1B loss, in cancer development. It also enables the discovery and testing of putative targeted therapeutic agents for DDEC with ARID1A/ARID1B dual deficiency.

## Supporting information

**S1 Table. Summary of somatic exonic non-synonymous SNVs and Indels in VOA1066 samples.** This table summarized the genomic changes and resulting amino acid changes of each somatic mutation as well as the allele frequency detected by exome sequencing. The functional impact of each SNV mutation were predicted by two softwares, SIFT and Polyphen 2. (XLSX)

**S1 Data. Source data for western blots.** The original western blots were included for Fig 4C, 4E and 4F. (PPTX)

## Acknowledgments

We appreciate the technical support by Winnie Yang, Cindy Shen and Zhengyang Zhou Xia.

## Author Contributions

**Conceptualization:** Yemin Wang, Jessica N. Mcalpine, C. Blake Gilks, David G. Huntsman.

**Data curation:** Yemin Wang.

**Formal analysis:** Yemin Wang, Martin Köbel, David Farnell.

**Funding acquisition:** Yemin Wang.

**Investigation:** Valerie Lan Tao, Chae Young Shin, Clara Salamanca, Shary Yuting Chen, Christine Chow, Susana Ben-Neriah.

**Project administration:** Yemin Wang.

**Resources:** Christian Steidl.

**Supervision:** Yemin Wang, David G. Huntsman.

**Writing – original draft:** Yemin Wang, Valerie Lan Tao.

**Writing – review & editing:** Yemin Wang, Martin Köbel, Christian Steidl, Jessica N. Mcalpine, C. Blake Gilks, David G. Huntsman.

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
