## [Decision Letter · Decision Letter 0]

6 Mar 2020

PONE-D-20-03388

Establishment and Characterization of VOA1066 cells: an Undifferentiated Endometrial Carcinoma Cell Line

PLOS ONE

Dear Dr. Wang,

Thank you for submitting your manuscript to PLOS ONE. After careful consideration, we feel that it has merit but does not fully meet PLOS ONE’s publication criteria as it currently stands. Therefore, we invite you to submit a revised version of the manuscript that addresses the points raised during the review process.

We would appreciate receiving your revised manuscript by Apr 20 2020 11:59PM. To enhance the reproducibility of your results, we recommend that if applicable you deposit your laboratory protocols in protocols.io, where a protocol can be assigned its own identifier (DOI) such that it can be cited independently in the future. For instructions see: http://journals.plos.org/plosone/s/submission-guidelines#loc-laboratory-protocols

We look forward to receiving your revised manuscript.

Kind regards,

Elizabeth Christie

Academic Editor

PLOS ONE

Journal Requirements:

2. Please provide additional details regarding participant consent. In the ethics statement in the Methods and online submission information, please ensure that you have specified what type of consent you obtained (for instance, written or verbal, and if verbal, how it was documented and witnessed).

3. At this time, we request that you  please report additional details in your Methods section regarding animal care, as per our editorial guidelines:

(1) Please state the source of the mice used in the study  

(2) Please provide details of animal welfare (e.g., shelter, food, water, environmental enrichment)

(3) Please describe any steps taken to minimize animal suffering and distress, such as by administering anaesthesia  

(4) Please include the method of euthanasia  

(5) Please describe the post-operative care received by the animals, including the specific clinical, physiological and behavioural criteria used to assess animal health and well-being for humane endpoints during the tumour xenograft study.

(6) Please state whether any of the mice died before meeting the criteria for humane endpoints.

Thank you for your attention to these requests.

4. At this time, we ask that you please provide the source, product number and any lot numbers of the inhibitors used in the drug sensitivity assays (Paclitaxel, Cisplatin and Tazemetostat).

5. To comply with PLOS ONE submission guidelines, in your Methods section, please provide additional information regarding your statistical analyses. For more information on PLOS ONE's expectations for statistical reporting, please see https://journals.plos.org/plosone/s/submission-guidelines.#loc-statistical-reporting.

7.  Thank you for stating the following in the Acknowledgments Section of your manuscript: 

"We appreciate the technical support by Winnie Yang, Cindy Shen and Zhengyang Zhou Xia. This work

was supported by research funds from the National Cancer Institute of NIH (United States)

(1R01CA195670-01), the Terry Fox Research Institute Initiative New Frontiers Program in Cancer

(TFF1021), the Canadian Institute of Health Research Foundation Grant (#154290), and the Carraresi

Foundation and the Sumiko Kobayashi Marks Memorial OVCARE Research Grants supported by the

VGH & UBC Hospital Foundation.".

"NO - Include this sentence at the end of your statement: The funders had no role in study design, data collection and analysis, decision to publish, or preparation of the manuscript.".

8. PLOS ONE now requires that authors provide the original uncropped and unadjusted images underlying all blot or gel results reported in a submission’s figures or Supporting Information files. This policy and the journal’s other requirements for blot/gel reporting and figure preparation are described in detail at https://journals.plos.org/plosone/s/figures#loc-blot-and-gel-reporting-requirements and https://journals.plos.org/plosone/s/figures#loc-preparing-figures-from-image-files. When you submit your revised manuscript, please ensure that your figures adhere fully to these guidelines and provide the original underlying images for all blot or gel data reported in your submission. See the following link for instructions on providing the original image data: https://journals.plos.org/plosone/s/figures#loc-original-images-for-blots-and-gels.

Additional Editor Comments (if provided):

Please note that certain details of animal experimental procedures are missing from the methods, particularly a description of the method of euthanasia, please update the methods accordingly.

Reviewers' comments:

Reviewer's Responses to Questions

**Comments to the Author**

1. Is the manuscript technically sound, and do the data support the conclusions?

Reviewer #1: Yes

Reviewer #2: Partly

2. Has the statistical analysis been performed appropriately and rigorously? 

Reviewer #1: Yes

Reviewer #2: N/A

3. Have the authors made all data underlying the findings in their manuscript fully available?

Reviewer #1: Yes

Reviewer #2: Yes

4. Is the manuscript presented in an intelligible fashion and written in standard English?

Reviewer #1: Yes

Reviewer #2: Yes

5. Review Comments to the Author

Reviewer #1: Wang et al. describe the generation of an undifferentiated human endometrial carcinoma cell line VOA1066, which harbours characteristics such as intact MMR, ER negativity, genomic stability and is capable of being grown as xenografts. This work is important given the highly aggressive nature of this type of endometrial cancer and the lack of models to investigate the disease. The cell line displays a truncated form of ARID1A, the loss of ARID1B and a p.D1810Y mutation in DICER1, which the authors determine likely retains wild type function. Given the model recapitulates the features of the original tumour from which it was derived, overall the study is well done and establishes a new experimental model that will facilitate future investigation of this rare endometrial cancer type. I have some comments for clarification and suggestions.

Specific points:

1. Figure 4D. Please provide details about the HEC50 cell line in the methods. Please explain the shift in molecular weight for SMARCA4 in the VOA1066 sample. As defects in ARID1A are often associated with constitutive PI3K/AKT pathway activation, the authors should assess pathway activation in addition to PTEN status.

2. The authors speculate that the truncated ARID1A protein will likely result in impaired incorporation into SWI/SNF complexes, but the claim that this model could be used as a model of both defective ARID1A and ARID1B would be improved if this were demonstrated.

3. In addition to the genomic characterisation of the VOA1066 cell line, RNA sequencing analysis would provide important insight into potential pathways that could be targeted clinically and bolster the utility of this line for mechanistic studies.

Typographical errors:

Ethics statement and page 5 - “eithics” should be “ethics.”

Page 9 – “alternations” should be “alterations.”

Reviewer #2: DDEC has recently been recognised as a distinct EC subtype for its aggressive behaviour. The mechanism associated with pathogenesis remains largely unknown. No matter how great proportion the lower-grade endometrioid adenocarcinoma accounted for the DDEC, the component of UEC contributed largely to a highly aggressive manner. Therefore, understating of DDEC starts with the knowledge of UEC. This paper established a UEC cell line and validate through specific genomic features. The study is of translational important but needs to address a couple of major points:

1. Molecular signature specific to UEC remains unclear. The cell line established in this paper could be useful to study to role of ARID1A and ARID1B mutations in UEC, but not quite useful for other inactivating mutations such as SMARCA4. Recent studies suggest a role for inactivation of SMARCA4 in UEC (Stewart et al, 2015; Karnezis et al, 2015) but according to the western blotting result of this study, SMARCA4 showed expression in the UEC cell line. It would also be useful to include IHC for SMARCA4.

2. Grade 2 or 3 endometrioid adenocarcinoma are the 2 most misdiagnoses of DDEC. More markers such as Mullerian epithelial markers: keratin, EMA and PAX8 should be included to validate the portion of UEC in the primary tumor as well as cell line to discriminate from grade 2 or 3 endometrioid adenocarcinoma.

3. The VOA1066 cell line had more than 30% protein altering mutations compared to the primary tumour, which could be problematic when understanding related mechanisms. It would be helpful to include a functional summary of the cell line specific mutations.

4. For the IHC of VOA1066 primary tumor, it would be reasonable to include positive controls for ER staining (such as from adjacent endometrioid portions).

6. PLOS authors have the option to publish the peer review history of their article (what does this mean?). If published, this will include your full peer review and any attached files.

Reviewer #1: Yes: Keefe Chan

Reviewer #2: No

---

## [Author Response · Author response to Decision Letter 0]

8 Sep 2020

Please refer to the "Responses to reviewers" document.

---

## [Decision Letter · Decision Letter 1]

28 Sep 2020

Establishment and Characterization of VOA1066 cells: an Undifferentiated Endometrial Carcinoma Cell Line

PONE-D-20-03388R1

Dear Dr. Wang,

We’re pleased to inform you that your manuscript has been judged scientifically suitable for publication and will be formally accepted for publication once it meets all outstanding technical requirements.

Kind regards,

Elizabeth Christie

Academic Editor

PLOS ONE

Additional Editor Comments (optional):

Reviewers' comments:

Reviewer's Responses to Questions

**Comments to the Author**

1. If the authors have adequately addressed your comments raised in a previous round of review and you feel that this manuscript is now acceptable for publication, you may indicate that here to bypass the “Comments to the Author” section, enter your conflict of interest statement in the “Confidential to Editor” section, and submit your "Accept" recommendation.

Reviewer #1: All comments have been addressed

Reviewer #2: All comments have been addressed

2. Is the manuscript technically sound, and do the data support the conclusions?

Reviewer #1: Yes

Reviewer #2: Yes

3. Has the statistical analysis been performed appropriately and rigorously? 

Reviewer #1: Yes

Reviewer #2: Yes

4. Have the authors made all data underlying the findings in their manuscript fully available?

Reviewer #1: Yes

Reviewer #2: Yes

5. Is the manuscript presented in an intelligible fashion and written in standard English?

Reviewer #1: Yes

Reviewer #2: Yes

6. Review Comments to the Author

Reviewer #1: The authors have addressed all of my comments from the initial review of the manuscript, which has been strengthened. The new model for UEC will be useful for researchers investigating this disease.

Reviewer #2: (No Response)

7. PLOS authors have the option to publish the peer review history of their article (what does this mean?). If published, this will include your full peer review and any attached files.

Reviewer #1: **Yes: **Keefe T Chan

Reviewer #2: No

---

## [Editor Report · Acceptance letter]

1 Oct 2020

PONE-D-20-03388R1 

Establishment and characterization of VOA1066 cells: an undifferentiated endometrial carcinoma cell line 

Dear Dr. Wang:

I'm pleased to inform you that your manuscript has been deemed suitable for publication in PLOS ONE. Congratulations! Your manuscript is now with our production department. 

Kind regards, 

on behalf of

Dr. Elizabeth Christie 

Academic Editor

PLOS ONE